# Conditional cash transfer program and child mortality: A cross-sectional analysis nested within the 100 Million Brazilian Cohort

**Dandara Ramos**[1,2☯]*, **Nívea B. da Silva**[1,3☯], **Maria Yury Ichihara**[1,2], **Rosemeire L. Fiaccone**[1,3], **Daniela Almeida**[1,4], **Samila Sena**[1], **Poliana Rebouças**[1,2], **Elzo Pereira Pinto Júnior**[1], **Enny S. Paixão**[1,5], **Sanni Ali**[1,5], **Laura C. Rodrigues**[1,5], **Maurício L. Barreto**[1,2]

**1** Center for Data and Knowledge Integration for Health (CIDACS), Fundação Oswaldo Cruz, Salvador, Bahia, Brazil, **2** Institute of Collective Health, Federal University of Bahia, Salvador, Bahia, Brazil, **3** Statistics Department, Institute of Mathematics and Statistics, Federal University of Bahia, Salvador, Bahia, Brazil, **4** Computer Science Department, Institute of Mathematics and Statistics, Federal University of Bahia, Salvador, Bahia, Brazil, **5** Faculty of Epidemiology and Population Health, London School of Hygiene & Tropical Medicine, London, United Kingdom

☯ These authors contributed equally to this work.
* dandara.ramos@ufba.br

**Data Availability Statement:** All data supporting the findings presented herein were obtained from Centro de Integração de Dados e Conhecimentos

## Abstract

### Background

Brazil has made great progress in reducing child mortality over the past decades, and a parcel of this achievement has been credited to the Bolsa Família program (BFP). We examined the association between being a BFP beneficiary and child mortality (1–4 years of age), also examining how this association differs by maternal race/skin color, gestational age at birth (term versus preterm), municipality income level, and index of quality of BFP management.

### Methods and findings

This is a cross-sectional analysis nested within the 100 Million Brazilian Cohort, a population-based cohort primarily built from Brazil's Unified Registry for Social Programs (Cadastro Único). We analyzed data from 6,309,366 children under 5 years of age whose families enrolled between 2006 and 2015. Through deterministic linkage with the BFP payroll datasets, and similarity linkage with the Brazilian Mortality Information System, 4,858,253 children were identified as beneficiaries (77%) and 1,451,113 (23%) were not. Our analysis consisted of a combination of kernel matching and weighted logistic regressions. After kernel matching, 5,308,989 (84.1%) children were included in the final weighted logistic analysis, with 4,107,920 (77.4%) of those being beneficiaries and 1,201,069 (22.6%) not, with a total of 14,897 linked deaths. Overall, BFP participation was associated with a reduction in child mortality (weighted odds ratio [OR] = 0.83; 95% CI: 0.79 to 0.88; $p < 0.001$). This association was stronger for preterm children (weighted OR = 0.78; 95% CI: 0.68 to 0.90; $p < 0.001$), children of Black mothers (weighted OR = 0.74; 95% CI: 0.57 to 0.97; $p < 0.001$),

para Saúde (CIDACS). Importantly, restrictions apply to access to these data, which contains sensitive information, were licensed for exclusive use in the current study and, due to privacy regulations from the Brazilian Ethics Committee are not openly available. Upon reasonable request and with express permission from CIDACS (mail to cidacs.curadoria@fiocruz.br) and approval from an ethical committee, controlled access to the data is possible. The dataset is registered under the following DOI handle: https://hdl.handle.net/20. 500.12196/CIDACS/65, which provides metadata and a register of all versions of the database.

**Funding:** This work was supported by the Grand Challenges Brazil grant MCTI/CNPq/MS/SCTIE/ Decit/Bill and Melinda Gates Foundation call N° 47/ 2014, (grant number opp1142172 awarded to the last author - "MLB"), and another grant by the Wellcome Trust (grant number 201912/B/16, also awarded to the last author - "MLB"). The funders had no role in study design, data collection and analysis, decision to publish, or preparation of the manuscript. URL of funders websites: https://www. gatesfoundation.org/ https://wellcome.org/.

**Competing interests:** All authors have completed the ICMJE uniform disclosure form at www.icmje. org/coi_disclosure.pdf and declare the financial support from the Grand Challenges Brazil grant, MCTI/CNPq/MS/SCTIE/Decit/Fundação Bill and Melinda Gates call N° 47/2014 and the Wellcome Trust (201912/B/16) for the submitted work. No financial relationships with any organizations that might have an interest in the submitted work in the previous three years; no other relationships or activities that could appear to have influenced the submitted work.

**Abbreviations:** BFP, Bolsa Família program; CadÚnico, Cadastro Único; CCT, conditional cash transfer; CIDACS, Center for Data and Knowledge Integration for Health; DMI, Decentralized Management Index; IPTW, inverse probability of treatment weighting; MHDI-R, Municipal Human Development Index–Renda; OR, odds ratio; PS, propensity score; SIM, Mortality Information System; SINASC, Live Birth Information System.

children living in municipalities in the lowest income quintile (first quintile of municipal income: weighted OR = 0.72; 95% CI: 0.62 to 0.82; $p < 0.001$), and municipalities with better index of BFP management (5th quintile of the Decentralized Management Index: weighted OR = 0.76; 95% CI: 0.66 to 0.88; $p < 0.001$). The main limitation of our methodology is that our propensity score approach does not account for possible unmeasured confounders. Furthermore, sensitivity analysis showed that loss of nameless death records before linkage may have resulted in overestimation of the associations between BFP participation and mortality, with loss of statistical significance in municipalities with greater losses of data and change in the direction of the association in municipalities with no losses.

## Conclusions

In this study, we observed a significant association between BFP participation and child mortality in children aged 1–4 years and found that this association was stronger for children living in municipalities in the lowest quintile of wealth, in municipalities with better index of program management, and also in preterm children and children of Black mothers. These findings reinforce the evidence that programs like BFP, already proven effective in poverty reduction, have a great potential to improve child health and survival. Subgroup analysis revealed heterogeneous results, useful for policy improvement and better targeting of BFP.

## Author summary

### Why was this study done?

- In 2019 alone, globally 5.2 million children died before reaching 5 years of age.

- Scientific research has pointed out conditional cash transfers as an effective policy to reduce child mortality rates, especially in low- and middle-income countries.

- No other study to our knowledge has analyzed the association between Brazil's Bolsa Família program (the world's largest conditional cash transfer program) and child mortality (ages 1–4 years).

### What did the researchers do and find?

- We analyzed a sample of 6,309,366 children under 5 years of age, between 2006 and 2015.

- Our study used methods to control for self-selection bias and explored results across subpopulations according to maternal education, maternal race/skin color, gestational age at birth, and municipal indicators of wealth and the quality of cash transfer management.

- We found a significant association between participating in the Bolsa Família program and a lower risk of mortality for children aged 1 to 4 years.

- This association was stronger for preterm children, children of Black mothers, children living in municipalities in the lowest income quintile, and children living in cities where the Bolsa Família program was best administered.

## What do these findings mean?

- Our findings are consistent with previous studies, pointing to a significant association between participation in conditional cash transfers and positive child health outcomes, such as lower child mortality.

- The greater association among children in more vulnerable situations suggests conditional cash transfers may help to promote equity, with stronger results among those in more need.

- In the future, other studies should aim to validate our findings by employing more rigorous impact evaluation techniques to support causal inference.

## Introduction

Worldwide, remarkable progress in reducing child mortality over the past 3 decades has been observed. From 1990 to 2018, the global under-5 mortality rate declined 58%, from 93 to 39 deaths per 1,000 live births. Still, the burden of child deaths remains significant: In 2019 alone, 5.2 million children died before their fifth birthday [1]. With an impressive 67% reduction in under-5 mortality from 1990 to 2015, Brazil met Millennium Development Goal 4 ahead of schedule [2], as rates dropped from 52 to 14 per 1,000 live births during this period, with an average annual decrease rate of 4.41%. However, if disaggregated from national levels, significant disparities can be found at both the beginning and end time points of this period, reflecting the still persistent inequalities in the country. In 1990, under-5 mortality varied from 38 to 114 per 1,000 live births in states in the North Region and Northeast Region, the poorest regions in the country, but ranged from 23 to 41 per 1,000 live births in the rest of Brazil. In 2015, these disparities lessened in magnitude but were still present, with rates ranging from 14 to 23 in states in the North Region and Northeast Region and 13 to 15 in the rest of the country [3].

   Given that a considerable proportion of child deaths are related to poverty and result from health issues that can be treated and prevented easily and economically [1,4], income redistribution initiatives have been a successful strategy for the improvement of child survival, especially in low- and middle-income countries [5–9]. Among such strategies are conditional cash transfer (CCT) programs, created with the purpose of breaking the intergenerational cycle of poverty by transferring cash to low-income families as long as they comply with the conditions of investing in their children's health and education [10]. Almost all countries in Latin America have CCT programs, and they are also present in countries including Bangladesh, Indonesia, Nigeria, Malawi, and Turkey [11,12], sparking interest even in high-income countries, given their impact on social development [13]. In Brazil, the CCT program, branded as the Bolsa Família program (BFP), was initiated in 2003 and rapidly implemented throughout the country, covering over 13 million families in 2015 and becoming the world's largest CCT.

Because of its extensive coverage and effectiveness in alleviating poverty, BFP has been pointed out as one of the driving forces behind the successful story of childhood mortality reduction in Brazil up to the year 2015 [14–16]. BFP's conditionality involves mainly health and education activities, as children must complete vaccine schedules and attend classes for at least 85% of the school year, and pregnant women must complete prenatal visits [17]. This is important for our understanding of the mechanisms through which the program can affect child mortality. As proposed by Rasella et al.'s model, BFP, like other CCTs, can affect child health through 2 pathways, one based on income improvement and the other on health-related requirements. Income improvement can objectively increase access to food and other living necessities, and health-related requirements can improve access to crucial services such as immunization, growth monitoring, and emergency care [15].

To date, there are 4 studies on BFP and child mortality [14–16,18], all pointing to a positive impact of the program on child survival. However, all of them are ecological and derived from aggregate municipal-level data. Here we tested the hypothesis that receiving a BFP stipend is associated with lower risk of child mortality, and we believe this study can contribute to the literature by using large-scale individual-level data from families enrolled in Brazil's Unified Registry for Social Programs (Cadastro Único [CadÚnico]), the BFP payroll database, and the Brazilian Mortality Information System (SIM), all linked as part of the 100 Million Brazilian Cohort [19,20].

Our main goals are (a) to analyze the association between receiving a BFP stipend and the risk of childhood mortality (1–4 years of age) and (b) to use subgroup analysis to explore whether the observed association varies according to indicators of poverty and quality of BFP management at the municipality level and according to maternal and perinatal characteristics, in particular, maternal race/skin color and gestational age at birth.

The rationale for pursuing the subgroup analysis by municipal-level poverty is based on previous research pointing to an inverse relationship between program coverage and municipal level of social and economic development [21,22]. Therefore, we expect the association between BFP participation and child mortality to be stronger in the poorest municipalities, since those have better targeting and coverage of the program. The basis for exploring stratification by municipal-level indicators of quality of BFP management is that there is evidence of considerable heterogeneity in these indicators among Brazilian municipalities [23–26], and that municipalities with better management and coverage of the program show stronger indicators of poverty reduction [27] and monitoring of program health conditionality [28]. We expect the association between BFP participation and child mortality to be stronger in municipalities with indicators of better CCT management, considering that there is greater monitoring of child health, immunization, and development in these contexts.

## Methods

### Data description

This is a cross-sectional analysis nested within the 100 Million Brazilian Cohort, a population-based cohort primarily built from CadÚnico, a shared registry for more than 20 social programs, which covers the poorest half of the Brazilian population (families with monthly income equal to or below 3 times minimum wage, approximately US$578). So far, the cohort includes data of approximately 114 million people (nearly half of the country's population) from 2001 to 2015, being continuously linked to other public health databases to generate data for population-health-relevant questions and epidemiological studies [19,20]. We restricted

our study window to the period 2006–2015 based on data linkage performance quality (S2 Text).

For the purposes of this study, the 100 Million Brazilian Cohort baseline dataset (consisting of data from individuals at their first registration in CadÚnico) was linked to 3 different databases: (a) the BFP payroll dataset, in order to identify the CCT's beneficiaries; (b) the Brazilian SIM, in order to identify the death records of children aged 1–4 in the 100 Million Brazilian Cohort baseline dataset; and (c) the Brazilian Live Birth Information System (SINASC), to assess relevant perinatal data such as birth weight and gestational age at birth.

**Data linkage.**   All the linkage steps were done at the individual level. The linkage process between the 100 Million Brazilian Cohort baseline dataset and BFP payroll dataset was deterministic, based on the NIS (Social Identification Number or "Número de Identificação Socia") number—a unique identifier similar to a social security number. The linkage between the 100 Million Brazilian Cohort baseline dataset, SIM, and SINASC was performed by similarity matching using CIDACS-RL, an open-source linkage algorithm from the Center for Data and Knowledge Integration for Health (CIDACS) that generates a similarity score on the basis of several identifiers [29]; the linked records were verified through manual analysis of a sample of 2,000 randomly selected pairs from all possible paired records (Table A in S2 Text).

The process to identify deaths of under-5 children linked to the 100 Million Brazilian Cohort was done in 3 stages, and a detailed report of the linkage methodology, including sensitivity and specificity indexes, can be found in S2 Text. Authors had access to a pseudonymized version of the linked database to create the study population, without the personally identifiable information fields.

## Statistical analyses

**Population definition.**   We included all children under 5 years of age whose families enrolled in CadÚnico between 1 January 2006 and 31 December 2015. To allow for the analysis of mortality between the ages 1 and 4 years, we included those who survived beyond the first year of life and up until age 5 until 31 December 2015, or died between the ages 1 and 4 years and 11 months during the same period of time. Other exclusions were related to inconsistencies between dates (e.g., date of registry in CadÚnico later than date of death) and unmeasured outcomes (Fig 1).

**Exposure definition.**   The beneficiary group (exposed) was defined as children whose family received a BFP stipend, uninterruptedly, from the first to the fifth year of the child's life. The non-beneficiary group (unexposed) was composed of children in families that did not receive a BFP stipend prior to the child reaching 5 years of age or death, receiving a stipend only after that time point or never receiving a stipend.

If BFP stipends were randomly assigned to families, one could assess the effect of the CCT on childhood mortality by simply comparing the difference in mortality between beneficiaries and non-beneficiaries. In fact, BFP is not randomly assigned. Instead, according to the program's eligibility criteria [17], it is a process of families' "self-selection" since whether a family receives the stipend or not is determined by per capita income and a set of family socioeconomic characteristics. Regarding per capita income, concerns about its reliability have been raised, given it is a self-reported measure [30]. To account for the issue of self-selection into the exposure group, we followed a kernel matching approach for the choice of a set of BFP non-beneficiary observations inside the CIDACS 100 Million Brazilian Cohort, allowing us to balance the 2 groups on their observable characteristics and control for possible bias arising from socioeconomic factors.

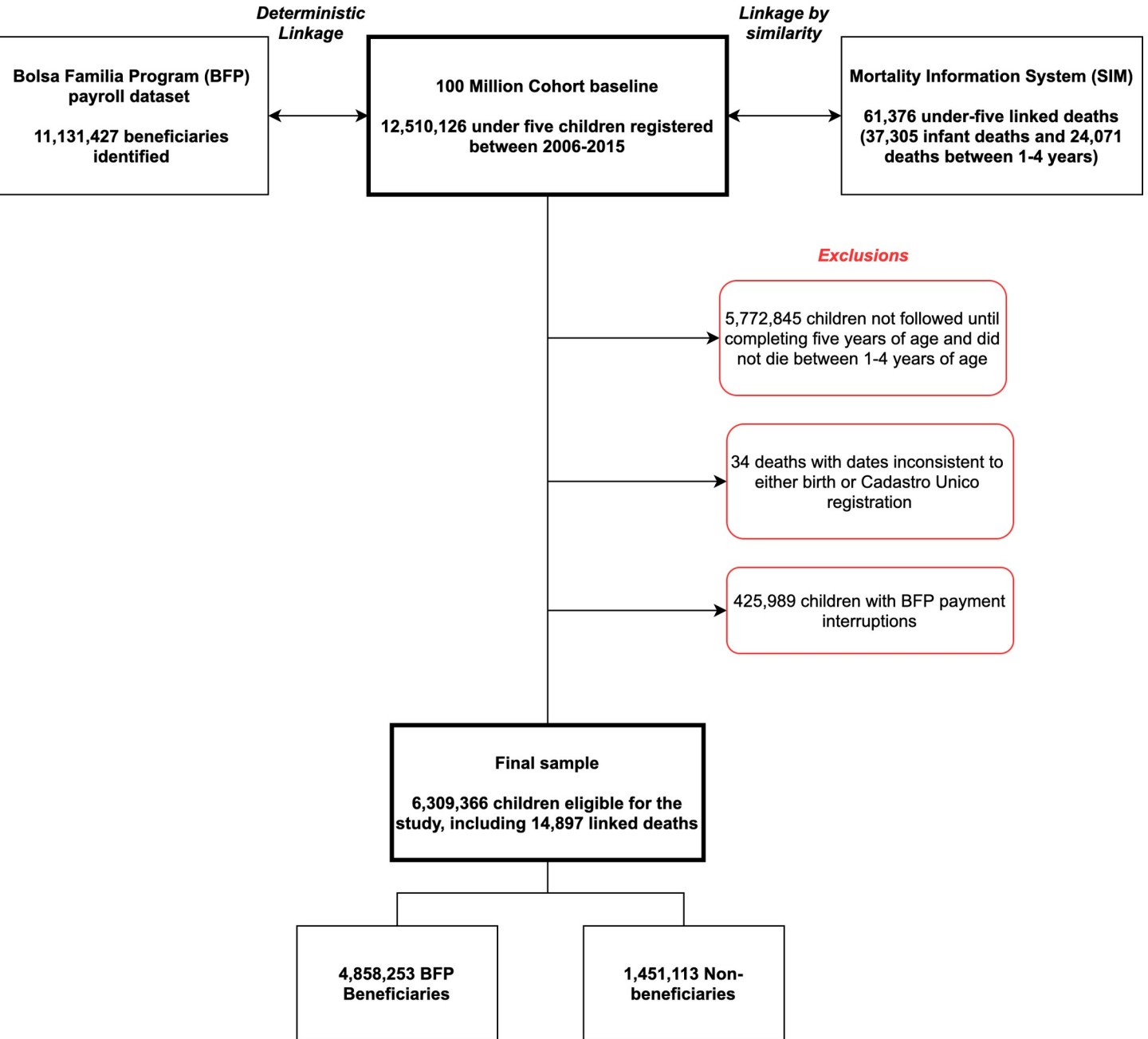

**Fig 1. Flowchart of selection of study population.** Flow diagram of selection and exclusion criteria for the population eligible for this study after linkage of the 100 Million Brazilian Cohort baseline dataset with Bolsa Família and mortality data.

The analysis plan originally outlined by the study protocol (S1 Text) anticipated the use of propensity score (PS) methods and regression discontinuity. Due to limitations of the income data in CadÚnico, we were not able to implement the latter.

**Analytical approach.** Our analysis consists of a combination of kernel matching and weighted logistic regressions. In the first part of our analysis, we used a logit model to estimate the probability of receiving a BFP stipend (PS) based on baseline characteristics of the child's mother and living conditions.

We include baseline characteristics of the child's mother and living conditions as regressors in the PS model using the following variables: household density ($\leq 2$ versus $>2$ people per room), maternal education (3 years of education or less, 4 to 7 years of education, or 8 years of education or more), self-reported maternal race/skin color (Black, white, Asian, mixed/brown [pardo], or indigenous), maternal marital status (lives with a partner [married or in a relationship] or no partner [single, widowed, or divorced]), maternal parity (0, 1, 2, 3, or 4 or more children), maternal age ($\leq 19$, 20–34, or $\geq 35$ years), region (North, Northeast, Central-West, Southeast, or South), and year of first registration in CadÚnico (ranging from 2006 to 2015). We chose not to include income as a PS predictor. Although BFP eligibility is, by regulation, defined by per capita income, it is a self-reported variable highly subject to manipulation [30]. Variables with more than 10% missing values were not included in the analysis.

After we estimated the PSs, we performed a kernel matching procedure, as proposed by Heckman et al. [31]. Kernel-based matching estimation is a non-parametric approach that uses weighted averages of all individuals in the control group to construct the counterfactual outcome, where more weight is given to units close to the one that needs to be matched. Intuitively, the procedure selects individuals who were not BFP beneficiaries but "look like" the set of beneficiary observations, in terms of their PSs, and gives them a greater weighting. Weights in kernel-based matching depend on the distance between each individual from the control group and the participant observation for which the counterfactual is estimated [32]. When applying this approach, one has to choose a kernel function and the bandwidth parameter, which determines how narrow a band of values around the participants' PSs receive high weights. In this work, we use an Epanechnikov kernel function with a bandwidth $h$, chosen through the pair-matching algorithm [33]. This yields a set of matching weights for the control group, which allows us to obtain an appropriate set of counterfactual observations. We impose a common support condition that drops beneficiary observations whose PS is higher than the maximum or less than the minimum of the non-beneficiaries.

To reduce bias, standard error estimates were obtained by bootstrap methodology, based on estimating from resampling with replacement from the original sample; this methodology has been widely applied to calculate standard error estimates in this setting [34–36].

In the final logistic models, weighted by the kernel weights from the previous stage, we adjust for relevant perinatal conditions (number of prenatal visits, birth weight, gestational age at birth, and type of delivery). For this method to provide unbiased estimates, we need to believe that—conditional on the sample restriction, common support condition, and kernel weights—the regressors are not correlated with the error term. If this assumption is satisfied, then our estimate represents an unbiased estimate of the "average treatment effect on the treated" (ATT). Besides this, for final logistic models, we also considered cluster effects on the estimation of standard errors, because the observations are clustered within a household, and adjusting for this cluster effect returns more robust estimates.

**Subgroup analysis.** The subgroup analysis was established by our research protocol (see S1 Text, objective 3). We aimed to explore BFP's association with child mortality across subgroups. To analyze the association between receiving a BFP stipend and child mortality according to individual characteristics, we looked at subgroups according to maternal race/skin color and gestational age at birth. To conduct this analysis, all kernel-weighted logistic models were calculated separately within each subgroup of gestational age at birth (preterm [<37 weeks] versus term [$\geq 37$ weeks]) and maternal race/skin color (Black, white, indigenous, mixed/brown). This implies that our subgroup analysis should be interpreted as "the estimates of $x$ on $y$ for group 1 and the estimates of $x$ on $y$ for group 2," as is appropriate with separate regressions.

We also analyzed heterogeneities in BFP's association with child mortality by presenting estimates of this relationship for the municipalities in the highest and lowest income quintiles and also those with different indexes of quality in BFP's management. To achieve this goal, we ranked municipalities into quintiles of per capita income (Municipal Human Development Index–Renda [MHDI-R]) and CadÚnico's Decentralized Management Index (DMI) and separately conducted all the analysis steps within each quintile of these indicators, including the estimation of PSs, kernel matching, and final weighted logistic models to provide separate estimates.

The DMI is an indicator of the quality of the management of BFP at the municipality level. It varies from 0 (worst) to 1 (best), and it is calculated based mainly on the timeliness of CadÚnico updates by the municipalities, their success in capturing families in extreme poverty, and their performance in monitoring the upholding of health and education conditions by the program's beneficiaries [37]. Given that in municipalities with better DMI beneficiaries are probably more compliant with the conditions and families in extreme poverty are more likely to be included in the program, we expect BFP to be associated with a greater reduction of child mortality in these contexts. Our choice to quintilize these municipal indicators represents an exploratory approach. Although there is theoretical support for selecting these variables [21–28], we did not identify any studies in the literature that have worked with these specific variables when analyzing BFP association with child health outcomes.

To formally assess whether the association between receiving a BFP stipend and child mortality varies across subgroups, we performed a statistical test for interaction by including BFP status × subgroup indicator terms in our subgroup models. Analysis of the presence of heterogeneity in BFP association with mortality across subgroups was based on the statistical significance ($p < 0.05$) of the interaction terms [38] and on applying the likelihood ratio test to evaluate the difference between nested models, considering that the model without the interaction term is nested within the model with the interaction term.

The use of interaction terms results in numerous hypothesis tests, especially in the context of categorical covariates. To account for that, we applied the Bonferroni correction and adjusted $p$-values for multiple comparisons within factor variable terms.

**Robustness checks.** We checked the robustness of our results by using inverse probability of treatment weighting (IPTW) as an alternative approach. For this analysis, the association between BFP participation and child mortality was estimated through logistic models with weights equal to PS/(1 − PS) for the non-beneficiaries and weights equal to 1 for the beneficiaries. Literature on the comparison between the properties of kernel matching and IPTW estimates suggests that IPTW surpasses kernel matching in terms of precision and, since it does not require the selection of a bandwidth parameter, is quicker to compute than kernel matching, reducing the time to bootstrap the standard errors [39].

Finally, to assess the impact of linkage bias in our estimates, we break down the analysis into 7 subgroups of municipalities according to their percentage of nameless records not submitted to linkage due to missing information (see S2 Text for more details). All analyses were done using Stata/MP version 15.0. This study is reported as per the REporting of studies Conducted using Observational Routinely-collected health Data (RECORD) guideline (S1 Checklist).

The study protocol was reviewed and approved by the Institute of Public Health Ethics Committee at the Federal University of Bahia (CAAE registration number: 56003716.0.3001.5030), and patient consent was not required as the study used only de-identified registry-based secondary data.

## Results

Of the 12,510,126 under-5 children registered in the 100 Million Brazilian Cohort between 2006 and 2015, 6,309,366 children from 4,627,984 families were initially included in this study (Fig 1). Among this sample, 4,858,253 were BFP beneficiaries (77%) and 1,451,113 (23%) were not. After the kernel matching procedure, 5,308,989 (84.1%) children were included in the final weighted logistic analysis, comprising 14,897 linked deaths. Regarding BFP status, 4,107,920 (77.4%) were beneficiaries and 1,201,069 (22.6%) were not. Table 1 shows the distribution of baseline characteristics between beneficiaries and non-beneficiaries, indicating that the 2 groups became more balanced in such covariates after the kernel procedure (S3 Text).

Regarding the association of BFP with child mortality, the weighted logistic regression analysis results are presented in Table 2. In the model adjusted for the number of prenatal visits, birth weight, gestational age at birth, and type of delivery, being in a family that received a BFP stipend was associated with lower mortality (weighted odds ratio [OR] = 0.83; 95% CI: 0.79 to 0.88; $p < 0.001$). This finding was consistent in the subgroup analyses for quintiles of municipal income, quintiles of DMI, maternal race/skin color, and gestational age at birth, with noticeable differences across levels of these indicators (Tables 3–6).

Across quintiles of municipal income, the association between BFP participation and mortality for children aged 1–4 years ranged from a 28% reduction in the odds of mortality in the poorest municipalities (weighted OR = 0.72; 95% CI: 0.62 to 0.82; $p < 0.001$) to a 13% reduction in the fourth quintile (weighted OR = 0.87; 95% CI: 0.76 to 0.98; $p < 0.001$) and no association in the fifth quintile, indicating a stronger association between BFP and lower risk of mortality for children living in the poorest 20% of municipalities of Brazil (Table 3).

The association between BFP participation and child mortality was also stronger in municipalities in the highest DMI quintile, ranging from a 12% reduction in the odds of mortality in the lowest quintile (weighted OR = 0.88; 95% CI: 0.81 to 0.96; $p < 0.001$) to a 24% reduction in the highest quintile (weighted OR = 0.76; 95% CI: 0.66 to 0.88; $p < 0.001$), suggesting a stronger association of BFP with lower risk of child mortality in municipalities in which the program is best administered (Table 4).

Considering differences across individual characteristics, an association between BFP participation and child mortality was found for all maternal race/skin color groups except for children of indigenous mothers (weighted OR = 0.99; 95% CI: 0.51 to 1.96; $p < 0.001$), possibly due to the relatively small sample size of this group. Estimates varied from a small 10% decrease in the odds of mortality for BFP beneficiary children of white mothers to a 19% decrease for BFP beneficiary children of mixed/brown (pardo) mothers (weighted OR = 0.81; 95% CI: 0.75 to 0.86; $p < 0.001$) and a 26% decrease for beneficiary children of Black mothers (Table 5). Considering gestational age at birth, an association between BFP participation and child mortality was found for both term (weighted OR = 0.84; 95% CI: 0.79 to 0.89; $p < 0.001$) and preterm (weighted OR = 0.78; 95% CI: 0.68 to 0.90; $p < 0.001$) children, with a stronger association for the latter (Table 6).

As a formal test of our subgroup effects, we conducted an interaction analysis (Tables F and G in S3 Text). The results of the likelihood ratio test indicated our interaction terms to be statistically significant for BFP status × MHDI-R quintile ($p < 0.001$), BFP status × DMI quintile ($p = 0.001$), BFP status × maternal race/skin color ($p = 0.002$), and BFP status × gestational age at birth ($p < 0.001$).

As illustrated in Fig 2, the predictive margins for probability of child mortality indicate noticeable heterogeneity across subgroups (all other variables held constant). The likelihood ratio test comparing the models with versus without the interaction terms was also significant in all 4 subgroup models (Tables F and G in S3 Text).

**Table 1. Baseline characteristics of Bolsa Família program (BFP) beneficiaries and non-beneficiaries before and after the kernel matching procedure—100 Million Brazilian Cohort, 2006 to 2015.**

| Characteristic | Absolute frequency and unweighted (crude) proportion | | | | | Kernel-weighted proportion | | |
|---|---|---|---|---|---|---|---|---|
| | BFP beneficiary (N = 4,858,253) | | Non-beneficiary (N = 1,451,113) | | Difference[1] | BFP beneficiary (N = 4,107,920) | Non-beneficiary (N = 1,201,069) | Difference[1] |
| | N | Percent | N | Percent | | | | |
| **Region** | | | | | | | | |
| North | 654,492 | 13.5 | 166,399 | 11.5 | 2.0 | 13.1 | 13.2 | −0.1 |
| Northeast | 1,970,529 | 40.6 | 413,395 | 28.5 | 12.1 | 38.2 | 37.7 | 0.6 |
| Southeast | 1,538,119 | 31.7 | 523,548 | 36.1 | −4.4 | 33.1 | 33.7 | −0.6 |
| South | 404,724 | 8.3 | 203,779 | 14.0 | −5.7 | 9.3 | 9.2 | 0.1 |
| Central-West | 290,389 | 6.0 | 143,992 | 9.9 | −3.9 | 6.2 | 6.3 | 0.0 |
| *Missing* | 0 | 0.0 | 0 | 0.0 | | | | |
| **Maternal education** | | | | | | | | |
| ≤3 years | 2,065,350 | 42.5 | 845,553 | 58.3 | −15.8 | 43.1 | 42.9 | 0.2 |
| 4–7 years | 1,924,239 | 39.6 | 446,234 | 30.7 | 8.9 | 40.9 | 41.2 | −0.3 |
| ≥8 years | 746,711 | 15.4 | 126,652 | 8.8 | 6.6 | 16.0 | 15.9 | 0.1 |
| *Missing* | 121,953 | 0.3 | 32,674 | 2.2 | −1.9 | | | |
| **Household density** | | | | | | | | |
| ≤2 people per room | 4,497,436 | 92.6 | 1,382,043 | 95.2 | −2.6 | 94.7 | 94.7 | 0.0 |
| >2 people per room | 247,166 | 5.1 | 23,589 | 1.6 | 3.5 | 5.3 | 5.3 | 0.0 |
| *Missing* | 113,651 | 0.2 | 45,481 | 3.1 | −2.9 | | | |
| **Maternal race/skin color** | | | | | | | | |
| White | 1,380,976 | 28.4 | 550,947 | 38.0 | −9.6 | 29.7 | 29.7 | 0.0 |
| Black | 236,240 | 4.9 | 56,641 | 3.9 | 1.0 | 4.7 | 4.9 | −0.2 |
| Asian descent[2] | 16,800 | 0.4 | 5,980 | 0.4 | 0.0 | 0.3 | 0.3 | 0.0 |
| Mixed/brown | 3,179,209 | 65.4 | 834,486 | 57.5 | 7.9 | 64.4 | 64.4 | 0.1 |
| Indigenous | 44,856 | 0.9 | 2,948 | 0.2 | 0.7 | 0.8 | 0.7 | 0.1 |
| *Missing* | 172 | 0.0 | 111 | 0.0 | | | | |
| **Maternal marital status** | | | | | | | | |
| Has a partner | 1,617,927 | 33.3 | 603,500 | 41.6 | −8.3 | 34.3 | 34.0 | 0.3 |
| No partner (single, divorced, widowed) | 3,149,217 | 64.8 | 822,818 | 56.7 | 8.1 | 65.7 | 66.0 | −0.3 |
| *Missing* | 91,109 | 0.2 | 24,795 | 1.7 | −1.5 | | | |
| **Maternal parity** | | | | | | | | |
| 0 children | 1,309,968 | 29.6 | 623,445 | 42.9 | −13.3 | 29.6 | 29.5 | 0.1 |
| 1 child | 1,332,313 | 27.4 | 412,560 | 28.4 | −1.0 | 30.1 | 30.7 | −0.6 |
| 2 children | 836,486 | 17.2 | 162,515 | 11.2 | 6.0 | 18.9 | 19.1 | −0.2 |
| 3 children | 438,084 | 0.9 | 56,445 | 3.9 | −3.0 | 9.9 | 9.8 | 0.1 |
| 4 children or more | 521,636 | 10.8 | 58,148 | 4.0 | 6.8 | 11.6 | 10.9 | 0.7 |
| *Missing* | 419,766 | 0.8 | 138,000 | 9.5 | −8.7 | | | |
| **Maternal age** | | | | | | | | |
| ≤19 years | 1,183,187 | 24.4 | 405,803 | 28.0 | −3.6 | 22.3 | 23.2 | −0.8 |
| 20–34 years | 3,355,218 | 69.1 | 962,022 | 66.3 | 2.8 | 70.8 | 70.4 | 0.4 |
| ≥35 years | 318,198 | 6.6 | 82,769 | 5.7 | 0.8 | 6.9 | 6.5 | 0.4 |
| *Missing* | 1,650 | 0.0 | 519 | 0.0 | | | | |

[1]The difference in proportions of each category between BFP beneficiaries and non-beneficiaries (BFP beneficiary proportion minus non-beneficiary proportion).

[2]The Asian group could not be included in the final models due to small sample size and number of linked deaths (n = 12).

**Table 2. Regression results: Coefficients of unadjusted and adjusted kernel-weighted logistic regressions of Bolsa Família Program (BFP) participation on mortality between ages 1 and 4 years.**

| Coefficient | Unadjusted model | | | Adjusted model[1] | | |
|---|---|---|---|---|---|---|
| | Weighted odds ratio (95% CI) | Robust standard error | *p*-Value | Weighted odds ratio (95% CI) | Robust standard error | *p*-Value |
| **Beneficiary status (BFP participation = 1)** | 0.84 (0.79 to 0.88) | 0.0232 | <0.001 | 0.83 (0.79 to 0.88) | 0.0231 | <0.001 |
| **Constant** | 0.0029 | 0.0001 | | 0.0024 | 0.0001 | |

Sample size after kernel matching = 5,308,989.

[1]Model adjusted for number of prenatal visits, birth weight, gestational age at birth, and type of delivery.

The robustness analysis with IPTW yielded similar results to the kernel-weighted analysis, without changes in the overall association between BFP participation and child mortality (Table J in S3 Text) or within subgroups (Tables K to N in S3 Text).

Although the linkage process yielded good overall indexes of sensitivity and specificity (Table A in S2 Text), the percentage of nameless death records not submitted to linkage at the municipal level varied considerably across Brazil. Supposing this loss was concentrated among BFP beneficiaries, this could lead to overestimation of our calculated association, and if concentrated among non-beneficiaries, this could lead to an underestimated association, as a parcel of what we considered non-cases in the non-beneficiary group could, in fact, be among the death records initially left out of linkage. To account for this possible bias and as a form of sensitivity analysis, we calculated the percentage of nameless death records submitted to linkage in each municipality, and categorized municipalities into 8 subgroups based on this percentage: 0% (all nameless death records excluded from linkage), 0.1% to 20.0%, 20.1% to 40.0%,

**Table 3. Regression results: Coefficients of adjusted kernel-weighted logistic regressions within subgroups of municipal quintiles of per capita income (Municipal Human Development Index–Renda [MHDI-R]).**

| MHDI-R | Weighted odds ratio[1] (95% CI) | Robust standard error | *p*-Value | N |
|---|---|---|---|---|
| **Model 3a** | | | | 713,577 |
| 1st quintile (lowest income) | 0.72 (0.62 to 0.82) | 0.050 | <0.001 | |
| Constant | 0.004 (0.004 to 0.005) | 0.0004 | <0.001 | |
| **Model 3b** | | | | 802,524 |
| 2nd quintile | 0.75 (0.66 to 0.85) | 0.047 | <0.001 | |
| Constant | 0.003 (0.003 to 0.004) | 0.0003 | <0.001 | |
| **Model 3c** | | | | 722,243 |
| 3rd quintile | 0.84 (0.73 to 0.97) | 0.062 | 0.020 | |
| Constant | 0.002 (0.002 to 0.003) | 0.0002 | <0.001 | |
| **Model 3d** | | | | 856,961 |
| 4th quintile | 0.87 (0.76 to 0.98) | 0.057 | 0.027 | |
| Constant | 0.002 (0.002 to 0.003) | 0.0002 | <0.001 | |
| **Model 3e** | | | | 2,210,567 |
| 5th quintile (highest income) | 0.92 (0.84 to 1.01) | 0.043 | 0.086 | |
| Constant | 0.002 (0.002 to 0.003) | 0.0002 | <0.001 | |

All the analytical steps (propensity score estimation, kernel matching, and weighted logistic regression) were conducted separately within each level of MHDI-R. All models (3a to 3e) were done separately, within each of the MHDI-R quintiles, and adjusted for prenatal visits, birth weight, gestational age at birth, and type of delivery. Unadjusted estimates are available in S3 Text.

[1]Beneficiary status (Bolsa Família participation = 1).

**Table 4. Regression results: Coefficients of adjusted kernel-weighted logistic regressions within subgroups of Cadastro Único's Decentralized Management Index (DMI).**

| DMI | Weighted odds ratio[1] (95% CI) | Robust standard error | p-Value | N |
|---|---|---|---|---|
| **Model 4a** | | | | 2,309,348 |
| 1st quintile (worst) | 0.88 (0.81 to 0.96) | 0.040 | 0.005 | |
| Constant | 0.002 (0.001 to 0.002) | 0.0001 | <0.001 | |
| **Model 4b** | | | | 1,008,954 |
| 2nd quintile | 0.88 (0.78 to 1.00) | 0.055 | 0.047 | |
| Constant | 0.002 (0.002 to 0.003) | 0.0002 | <0.001 | |
| **Model 4c** | | | | 689,239 |
| 3rd quintile | 0.83 (0.72 to 0.96) | 0.061 | 0.010 | |
| Constant | 0.003 (0.002 to 0.003) | 0.0002 | <0.001 | |
| **Model 4d** | | | | 675,491 |
| 4th quintile | 0.79 (0.69 to 0.91) | 0.056 | 0.001 | |
| Constant | 0.002 (0.001 to 0.002) | 0.0001 | <0.001 | |
| **Model 4e** | | | | 622,812 |
| 5th quintile (best) | 0.76 (0.66 to 0.88) | 0.055 | <0.001 | |
| Constant | 0.003 (0.002 to 0.003) | 0.0003 | <0.001 | |

All the analytical steps (propensity score estimation, kernel matching, and weighted logistic regression) were conducted separately within each level of DMI. All models (4a to 4e) were done separately, within each of the DMI quintiles, and adjusted for prenatal visits, birth weight, gestational age at birth, and type of delivery. Unadjusted estimates are available in S3 Text.

[1]Beneficiary status (Bolsa Família participation = 1).

40.1% to 60.0%, 60.1% to 80.0%, 80.1% to 90.0%, 90.1% to 99.9%, and 100.0% (all nameless death records submitted to linkage) (Table C in S2 Text).

Analyzing the association between BFP participation and mortality in children aged 1–4 years in each of these subgroups, we found that in the group of municipalities without any loss of nameless records from linkage, the measured association changed direction, although the

**Table 5. Regression results: Coefficients of adjusted kernel-weighted logistic regressions within subgroups of maternal race/skin color.**

| Maternal race/skin color | Weighted odds ratio[1] (95% CI) | Robust standard error | p-Value | N |
|---|---|---|---|---|
| **Model 5a** | | | | 1,701,111 |
| White | 0.90 (0.83 to 0.99) | 0.041 | 0.019 | |
| Constant | 0.002 (0.002 to 0.002) | 0.0001 | <0.001 | |
| **Model 5b** | | | | 3,311,091 |
| Mixed/brown (pardo) | 0.81 (0.75 to 0.86) | 0.028 | <0.001 | |
| Constant | 0.003 (0.002 to 0.003) | 0.0001 | <0.001 | |
| **Model 5c** | | | | 239,587 |
| Black | 0.74 (0.57 to 0.97) | 0.101 | 0.029 | |
| Constant | 0.002 (0.002 to 0.003) | 0.0004 | <0.001 | |
| **Model 5d** | | | | 35,690 |
| Indigenous | 0.99 (0.51 to 1.96) | 0.345 | 0.993 | |
| Constant | 0.008 (0.004 to 0.021) | 0.004 | <0.001 | |

All the analytical steps (propensity score estimation, kernel matching, and weighted logistic regression) were conducted separately within each level of maternal race/skin color. All models (5a to 5d) were done separately, within each of these levels, and adjusted for prenatal visits, birth weight, gestational age at birth, and type of delivery. Unadjusted estimates are available in S3 Text.

[1]Beneficiary status (Bolsa Família participation = 1).

**Table 6. Regression results: Coefficients of adjusted kernel-weighted logistic regressions within subgroups of gestational age at birth.**

| Gestational age at birth | Weighted odds ratio[1] (95% CI) | Robust standard error | *p*-Value | N |
|---|---|---|---|---|
| **Model 6a** | | | | 4,960,905 |
| ≥37 weeks | 0.84 (0.79 to 0.89) | 0.025 | <0.001 | |
| Constant | 0.002 (0.002 to 0.003) | 0.001 | <0.001 | |
| **Model 6b** | | | | 345,266 |
| <37 weeks | 0.78 (0.68 to 0.90) | 0.057 | <0.001 | |
| Constant | 0.004 (0.003 to 0.005) | 0.004 | <0.001 | |

All the analytical steps (propensity score estimation, kernel matching, and weighted logistic regression) were conducted separately within each level of gestational age at birth. All models (6a and 6b) were done separately, within each of these levels, and adjusted for prenatal visits, birth weight, and type of delivery. Unadjusted estimates are available in S3 Text.

[1]Beneficiary status (Bolsa Família participation = 1).

lower bound of the confidence interval was close to the null value. For the subgroups of municipalities with 40.0% or less of their nameless records submitted, and those with 90.1%–99.9% submitted, the null value is well within their confidence intervals, indicating no significant association between BFP participation and mortality in these subgroups (Table C in S2 Text).

## Discussion

We found that receiving a BFP stipend is significantly associated with a reduced probability of death between the ages 1 and 4 years. This association was stronger for preterm children, children of Black mothers, children living in poorer municipalities, and children living in municipalities with better indexes of BFP management. These findings are consistent with previous studies reporting significant effects of BFP [14–16,18] and other CCTs on childhood mortality [7,40,41]. However, while these previous studies used aggregate municipal data, the present one used individual data from the 100 Million Brazilian Cohort [19,20].

Issues regarding inference based on ecological versus individual-level studies have been discussed for decades in various fields of research. We believe that interpreting individual-level findings is an advantage and a contribution of the present study to the knowledge on BFP, since we were able to deal with potential confounders through adjusting for individual variables, and using weighting techniques to address selection bias and covariates that are known predictors of child mortality.

Because of our large sample, it was also possible to explore heterogeneity in the association between BFP participation and mortality across multiple subgroups, findings that, when taken as a whole, are aligned with a recent review on CCTs and child health in low- and middle-income countries showing that such programs show substantial heterogeneity across subgroups defined not only geographically but also by indicators of socioeconomic status and program implementation intensity [42].

Considering the subgroups of maternal race/skin color and gestational age at birth—2 important upstream determinants of child health—our results went in a similar direction. The strongest association of the cash transfer with reduced child mortality was found among Black individuals, a group that is historically underprivileged socially and economically and in access to health in Brazil [43,44]. Studies have suggested that women exposed to a CCT program engage in higher maternal and child health service utilization and show substantial schooling accumulation. Both healthcare utilization and maternal education are significant predictors of child mortality, especially among preterm babies, which could explain the stronger association between the CCT program and child mortality among preterm babies. As proposed by Cooper

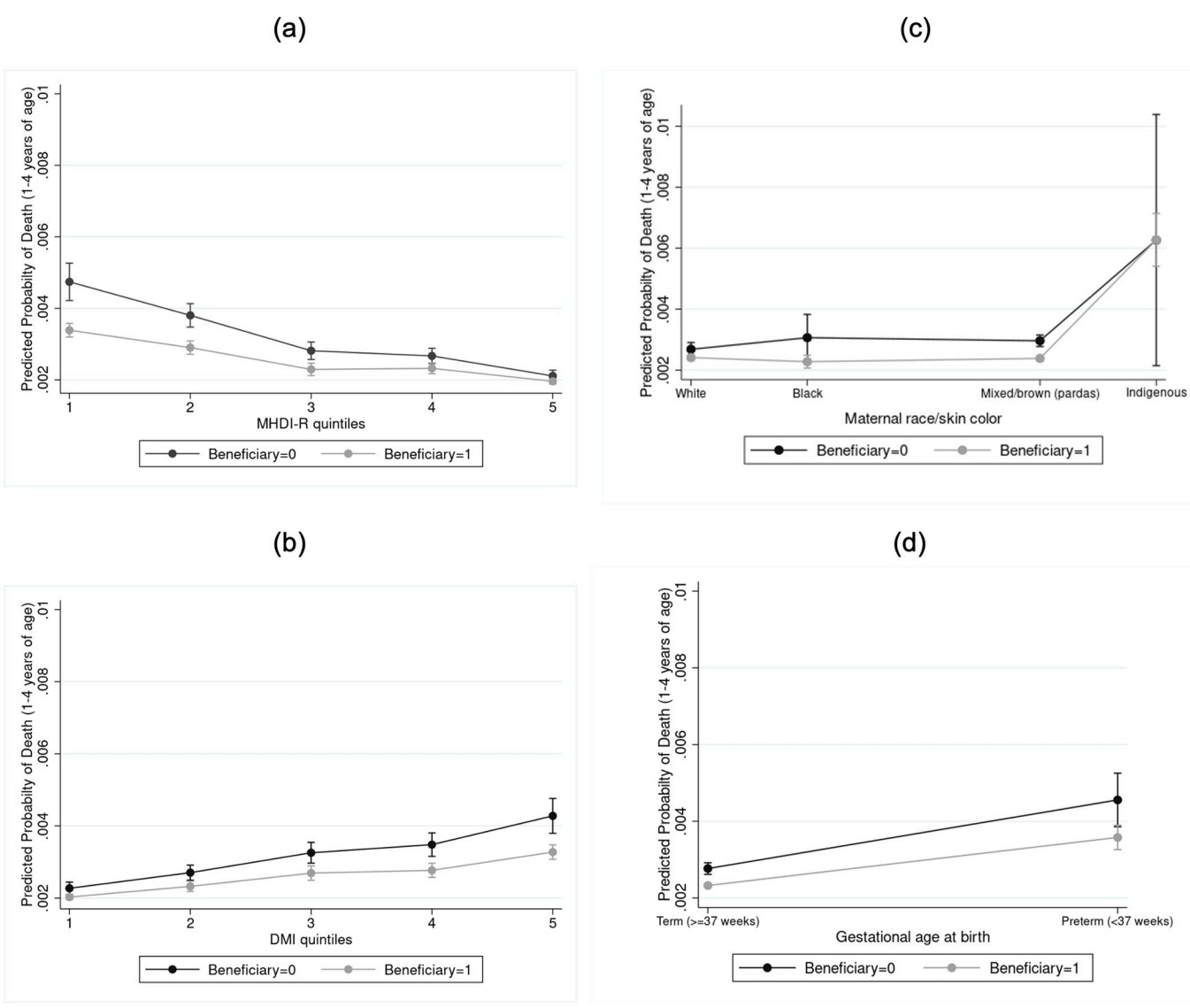

**Fig 2. Predictive margins for probability of child mortality, with 95% confidence intervals, by subgroup.** Beneficiary status: 0 = non-beneficiary; 1 = Bolsa Família program beneficiary. (a) Predictive margins (95% CI) by quintile of municipal per capita income (Municipal Human Development Index–Renda [MHDI-R]). Model adjusted for number of prenatal visits, birth weight, gestational age at birth, and type of delivery. (b) Predictive margins (95% CI) by quintile of Cadastro Único Decentralized Management Index (DMI). Model adjusted for number of prenatal visits, birth weight, gestational age at birth, and type of delivery. (c) Predictive margins (95% CI) by maternal race/skin color. Model adjusted for number of prenatal visits, birth weight, gestational age at birth, and type of delivery. (d) Predictive margins (95% CI) by gestational age at birth. Model adjusted for number of prenatal visits, birth weight, and type of delivery.

et al.'s recent review of the heterogeneous effects of CCTs, this suggests that cash transfers might work to augment or attenuate upstream determinants of child health [42].

In addition, we found evidence of heterogeneous results regarding BFP and mortality between the ages 1–4 years across broader contextual indicators, such as quintiles of municipal per capita income and DMI, an index that measures the quality of BFP and CadÚnico administration. In accordance with our hypothesis, the association between BFP participation and child mortality was stronger for beneficiaries living in poorer municipalities, meaning these cities' beneficiary children had lower odds of mortality than their non-beneficiary counterparts.

The relationship between socioeconomic status and health is one of the most robust and well-documented findings in social and health sciences. However, findings regarding the differential impacts of cash transfers across contexts of different poverty levels are not conclusive, with effect gradients going in opposite directions in studies of CCTs and child health conducted in Mexico, Ecuador, Niger, and India [45–49].

Regarding heterogeneous results across DMI levels, our results align with previous studies finding CCT effects to be dependent on implementation quality and management indicators [50–52]. As the 100 Million Brazilian Cohort comprises the poorest half of the country, this finding is of great importance, showing that not only the presence but also the quality of poverty-alleviating policies matters when targeting health promotion among vulnerable populations. This finding is aligned with our stated hypothesis that BFP's association with child mortality would be stronger in contexts where the program was best administered.

## Strengths and weaknesses of the study

Our study has notable strengths. We used individual data from a population-level database of more than 6 million children, making this the largest study on CCTs and child mortality to date, to the best of our knowledge. We applied a robust analytical approach, using kernel-based PS weighting to account for measured socioeconomic confounders and verified our results through an alternative approach, using an inverse probability weighted analysis. The beneficiary and non-beneficiary groups were well balanced for distributions of covariates, and weighted and stratified analyses confirmed our main findings to be consistent across several subgroups. We have also accounted for heterogeneities not previously assessed regarding the association of BFP participation with child health outcomes, providing specific results by maternal race/skin color group, gestational age at birth, and municipal level of poverty and program administration quality.

Through the kernel-matching procedure, we limited the potential for measured confounders to influence the result. Nevertheless, important unmeasured factors need to be considered, especially family income, a variable that could not be included in this study given its possibility for manipulation because it is a self-reported measure that can influence eligibility. Therefore, our methodology's main limitation is that our PS approach does not account for this and other possible unmeasured confounders. This could be dealt with by an instrumental variable approach, but we were unable to identify a suitable instrument in the data that is currently available within the 100 Million Brazilian Cohort. Future studies should revisit our hypothesis through a more rigorous quasi-experimental approach in order to support causal inference.

Since our evidence is associational, we recognize that causal claims should not be made solely based on our findings. Approaches such as the ones applied by Okeke and Abubakar to a Nigerian CCT [12] and Barham to Mexico's Progresa CCT [9] through randomized experiments are closer to providing all the elements expected for causal inference and impact evaluation. In the context of BFP, however, such experimental designs are not viable due to the program's characteristics and data availability. Nonetheless, we believe our contribution is still valuable to the broader literature on CCTs and child health by providing evidence based on a large population and focusing on several subgroups.

Limitations also arise from the linkage process, as loss of nameless records before linkage may have resulted in a possible overestimation of the association between program participation and child mortality, since we observed that in the municipalities without such losses, the direction of the measured association changed, and the lower bound of the confidence interval was very close to the null value of 1.00. Furthermore, in the very few municipalities ($n = 22$) where losses were greater than 60%, the association between BFP participation and mortality lost statistical significance.

Considering, however, the distribution of our study population across these municipal strata of nameless records lost before linkage, over 70% of our observations came from municipalities where the measured association was consistent with the one reported for the whole sample, with kernel-weighted odds ratios varying between 0.77 and 0.85 (Table C in S2 Text). It is also important to consider that nameless death records correspond to only 3.99% of the total death records for children aged 1–4 years in Brazil in our analysis period (5,218 nameless records out of a total of 130,858 deaths; Table B in S2 Text), and that less than 1% (22 municipalities) of the Brazilian municipalities had losses of linkage records greater than 40%.

Lastly, we were not able to isolate the effect of other interventions that could also be targeting poor families with children inside CadÚnico, such as the Child Labor Eradication Program (PETI) and the housing program (Minha Casa Minha Vida).

## Conclusions

CCTs like BFP have a great potential to improve child health in vulnerable populations. Using data from the 100 Million Brazilian Cohort, we conclude that BFP participation had a significant association with lower risk of child mortality, especially in municipalities with low per capita income. Moreover, as the magnitude of the association was greater in municipalities with better indicators of BFP targeting and management, we can also conclude that when properly administered, these programs can reinforce their relevance in improving child health outcomes.

Our findings were mostly consistent with other studies that have reported an important positive impact of BFP and other similar CCTs. Furthermore, this was, to our knowledge, the first evaluation of the association between BFP participation and child mortality based on individual-level data, and the first to account for heterogeneity, providing evidence necessary for better targeting and for generating more precise knowledge in this regard.

## Supporting information

**S1 Checklist. RECORD Checklist.** REporting of studies Conducted using Observational Routinely-collected health Data (RECORD) guideline.
(DOCX)

**S1 Fig. Linkage process between the CIDACS 100 Million Brazilian Cohort, mortality, and live birth records.** Flowchart of the linkage process between the CIDACS 100 Million Brazilian Cohort baseline dataset, mortality data (Brazilian Mortality Information System [SIM]), and live birth records (Brazilian Live Birth Information System [SINASC]).
(TIF)

**S2 Fig. Receiver operating characteristic (ROC) curve for the linkage between deaths of children aged 1–4 years (with name on the death certificate) and the CIDACS 100 Million Brazilian Cohort baseline dataset.**
(TIF)

**S3 Fig. Receiver operating characteristic (ROC) curve for the linkage between deaths of children under 5 years (without name on the death certificate—using the name of the mother) and the CIDACS 100 Million Brazilian Cohort baseline dataset.**
(TIF)

**S4 Fig. Propensity scores common support area.** Distribution of propensity scores across beneficiaries and non-beneficiaries.
(TIF)

**S1 Text. Research protocol.**
(PDF)

**S2 Text. Data linkage information.**
(DOCX)

**S3 Text. Kernel-matching procedure and robustness check.** Information on propensity
score estimation, kernel matching, subgroup analysis, and robustness check.
(DOCX)

## Author Contributions

**Conceptualization:** Dandara Ramos, Maria Yury Ichihara, Rosemeire L. Fiaccone, Poliana
Rebouças, Laura C. Rodrigues, Maurício L. Barreto.

**Data curation:** Maria Yury Ichihara, Rosemeire L. Fiaccone, Daniela Almeida, Samila Sena.

**Formal analysis:** Dandara Ramos, Nívea B. da Silva, Rosemeire L. Fiaccone.

**Funding acquisition:** Maria Yury Ichihara, Rosemeire L. Fiaccone, Laura C. Rodrigues, Maurício L. Barreto.

**Investigation:** Dandara Ramos, Nívea B. da Silva, Maria Yury Ichihara, Maurício L. Barreto.

**Methodology:** Dandara Ramos, Nívea B. da Silva, Rosemeire L. Fiaccone, Daniela Almeida,
Samila Sena, Maurício L. Barreto.

**Project administration:** Maria Yury Ichihara, Maurício L. Barreto.

**Resources:** Maria Yury Ichihara, Maurício L. Barreto.

**Supervision:** Maria Yury Ichihara, Rosemeire L. Fiaccone, Maurício L. Barreto.

**Validation:** Daniela Almeida, Samila Sena, Maurício L. Barreto.

**Visualization:** Daniela Almeida.

**Writing – original draft:** Dandara Ramos, Nívea B. da Silva, Daniela Almeida, Samila Sena,
Elzo Pereira Pinto Júnior, Enny S. Paixão, Maurício L. Barreto.

**Writing – review & editing:** Dandara Ramos, Nívea B. da Silva, Maria Yury Ichihara, Rosemeire L. Fiaccone, Daniela Almeida, Samila Sena, Poliana Rebouças, Elzo Pereira Pinto
Júnior, Enny S. Paixão, Sanni Ali, Laura C. Rodrigues, Maurício L. Barreto.

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
