## [Editor Report · Decision Letter 0]

9 Dec 2020

Dear Dr Ramos, 

Thank you for submitting your manuscript entitled "Conditional cash transfer program and child mortality: a study of the 100 Million Brazilian Cohort" for consideration by PLOS Medicine.

Your manuscript has now been evaluated by the PLOS Medicine editorial staff, as well as by the special issue guest editors, and I am writing to let you know that we would like to send your submission out for external peer review.

Kind regards,

Artur A. Arikainen,

Associate Editor

PLOS Medicine

---

## [Decision Letter · Decision Letter 1]

22 Feb 2021

Dear Dr. Ramos,

Thank you very much for submitting your manuscript "Conditional cash transfer program and child mortality: a study of the 100 Million Brazilian Cohort" (PMEDICINE-D-20-05781R1) for consideration in PLOS Medicine’s Special Issue on Global Child Health.

Your paper was evaluated by a senior editor and discussed among all the editors here. It was also discussed with the Special Issue Guest Editors, and sent to independent reviewers, including a statistical reviewer. The reviews are appended at the bottom of this email and any accompanying reviewer attachments can be seen via the link below:

[LINK]

In light of these reviews, I am afraid that we will not be able to accept the manuscript for publication in the journal in its current form, but we would like to consider a revised version that addresses the reviewers' and editors' comments. Obviously we cannot make any decision about publication until we have seen the revised manuscript and your response, and we plan to seek re-review by one or more of the reviewers. 

We expect to receive your revised manuscript by Mar 15 2021 11:59PM. Please email us (plosmedicine@plos.org) if you have any questions or concerns.

We look forward to receiving your revised manuscript. 

Sincerely,

Caitlin Moyer, Ph.D.

Associate Editor 

PLOS Medicine

plosmedicine.org

1.Title: Please revise your title according to PLOS Medicine's style. Your title must be nondeclarative and not a question. It should begin with main concept if possible. "Effect of" should be used only if causality can be inferred, i.e., for an RCT. Please place the study design ("A randomized controlled trial," "A retrospective study," "A modelling study," etc.) in the subtitle (ie, after a colon).

2.Data availability statement: The Data Availability Statement (DAS) requires revision. Please include the relevant information for requesting access to the data from CIDACS (weblink and/or email contact). The contact cannot be one of the authors of the study. It is not clear if the relevant data are available by accessing the provided link.

3.Throughout: Please include line numbers with the revised version of the manuscript.

4.Throughout: Please use square brackets for in-text citations, like this [1].

5.Throughout: Please use “White” rather than “Caucasian”

6.Abstract: Please structure your abstract using the PLOS Medicine headings (Background, Methods and Findings, Conclusions).

7.Abstract Background: Provide the context of why the study is important. The final sentence should clearly state the study question.

8.Abstract: Methods and Findings: Please be sure the abstract describes the population and setting, number of participants, years during which the study took place and main outcome measures.

9. Abstract: Methods and Findings: Please quantify the main results (with 95% CIs and p values).

10. Abstract: Methods and Findings: In the last sentence of the Abstract Methods and Findings section, please describe the main limitation(s) of the study's methodology.

11. Abstract: Conclusions: Please address the study implications without overreaching what can be concluded from the data; the phrase "In this study, we observed ..." may be useful.

12. Author Summary: At this stage, we ask that you include a short, non-technical Author Summary of your research to make findings accessible to a wide audience that includes both scientists and non-scientists. The Author Summary should immediately follow the Abstract in your revised manuscript. This text is subject to editorial change and should be distinct from the scientific abstract. Please see our author guidelines for more information: https://journals.plos.org/plosmedicine/s/revising-your-manuscript#loc-author-summary

13. Introduction: Please temper claims of primacy: e.g in second last paragraph of introduction: “To the best of our knowledge, this is the first study using nationwide individual level data to analyze the association of BFP with child mortality.”

14. Methods: Thank you for including the RECORD checklist. Please add the following statement, or similar, to the Methods: "This study is reported as per the REporting of studies Conducted using Observational Routinely-collected Data (RECORD) guideline (S1 Checklist)."

15. Methods: Did your study have a prospective protocol or analysis plan? Please state this (either way) early in the Methods section.

16. Methods: Please move the information pertaining to the study ethical approval and participant consent to the Methods section.

17. Throughout: Please clarify the language used- perhaps “highest income” or “greatest wealth” rather than “richest”

18. Results: For all findings presented in the text, please provide both the 95% CIs and p values along with the odds ratios. For the adjusted ORs, please also provide results from unadjusted analyses, this can be in a supporting information table.

19. Results: Please avoid the use of causal language (e.g. “effect”) by revising to the following: “Across quintiles of municipal income, associations with BFP ranged from a 28% reduction in the odds of mortality in the poorest municipalities (weighted OR= 0.72; CI 95% [0.62-0.82]) to 13% (weighted OR= 0.87; CI 95% [0.76-0.98]) in the fourth quintile, with no statistically significant association identified with the fifth quintile, indicating a stronger association between BFP and lower risk of mortality for children living in the poorest 20% municipalities of Brazil compared to those in the highest quintile (richest).” Please clarify whether comparisons were done between quintiles, to support the second half of the statement.

20. Results: Please clarify whether tests were done comparing DMI quintiles: “The association between BFP and mortality between ages 1-4 was also stronger in municipalities at the highest DMI quintile, ranging from a 12% reduction in the odds of mortality in the lowest quintile (weighted OR=0.88; CI 95% [0.81-0.96]) to 24% in the highest (weighted OR=0.76; CI 95% [0.66-0.88]), suggesting a stronger association of BFP with lower risk of child mortality in municipalities in which the program is best administered.”

21. Results: Please revise to avoid causal language: “Considering differences in association across individual characteristics, associations between BFP and child mortality were found in all race groups…” or similar.

22. Results: Please provide the analyses supporting this statement, or clarify if statistical significance is meant here: “The percentage of nameless death records not submitted to linkage at the municipal

level were significantly different across Brazil.”

23. Results: Please clarify this sentence, revising to avoid causal language: “The null value for the 95%

and 90% subgroups of municipalities are well within their confidence intervals, indicating no significant effects of the program in these subgroups (see Table 1c - supplementary material 1).”

24. Discussion: Please avoid the use of causal language throughout, e.g.”effect on” in the second paragraph: “Because of our large sample, it was also possible to explore heterogeneity in effect of BFP on 1-4 years old mortality across multiple subgroups…”

25. Conclusions: Please revise to avoid the use of causal language “Using data from the 100 Million Brazilian Cohort, we conclude that the programme had a significant effect on reducing child mortality, especially in municipalities with low per capita income. Moreover, as the magnitude of the effect grew in municipalities

with better indicators of BFP targeting and management, we can also conclude that when properly administered these programmes can increase their already significant impact on child health outcomes.”

26. References: Please use the "Vancouver" style for reference formatting, and see our website for other reference guidelines https://journals.plos.org/plosmedicine/s/submission-guidelines#loc-references

27. Checklist: When completing the RECORD/STROBE items on the checklist, please use section and paragraph numbers, rather than page numbers.

28. Figures/Tables: Please provide titles and legends for all figures and tables.

29. Figure 1: Please provide a descriptive legend, and please define the abbreviation for PBF.

30. Figure 2: Please indicate in the legend if these are adjusted or unadjusted OR- and if adjusted please list the variables included in the adjustment. Please provide the unadjusted analyses.

31. Table 1: Please define abbreviations in the legend (BFP) and please describe the BFP1-BFP0 in the legend.

32. Table 2 and Table 3: Please provide the p values associated with these OR

33. Table 3: Please indicate in the legend if these are adjusted OR, and indicate the variables included in the adjustment in the legend.

34. Supporting information files: Please include a descriptive title and legend for each figure/table in the Supporting Information files.

35. Supporting information Table 1C: Please revise the title to avoid causal implications: “Table 1c - Estimates of BFP impact on child mortality…” Please also provide p values, and note factors included for adjustment.

36. Supporting information Table 2A: Please also provide p values for these associations.

Comments from the reviewers:

Reviewer #1: It is a relevant manuscript to the public health area, since it addresses the issue related to the access to cash transfer program for low-income Brazilian population and infant mortality. Therefore, it is aligned with the scope of the journal Plos Medicine. 

The manuscript aimed to investigate the association between receiving Programa Bolsa Familia and the risk of mortality before 5 years old, through a 100 Million Brazilian Cohort whose data were obtained from Cadastro Único (CadÚnico). 

The manuscript is well written and structured. 

In the Introduction section, the authors described the present state of cientific literature related to the central theme, pointing out how far the studies have come and the gap to be filled to advance knowledge about the effects of income transfer programs in reducing poverty and mortality of children under 05 years. In this sense, the hypothesis was highlighted.

In the Methods section, procedures and analyzes for achieving the objectives were consistently listed. Results and Discussion were equally clear and concise, pointing out the reach and the limits of the study. In the Conclusion section, authors summarized th fundamental findings. 

Due to its unprecedented nature, as it is the first national study with individual data investigating the effects of a cash transfer program on infant mortality, I recommend the manuscript to be eligible for publication in Plos Medicine Journal, after minor changes listed below:

Abstract: 

"Conclusions: These findings reinforce the evidence that CCTs like BFP..."

- I suggest the authors to write the full name for CCTs, once it is the first time it appears in the text.

Introduction:

- Fourth line: authors might write the full name of MGD 4

Tables and Figures: 

- It is necessary to review the titles of Tables and Figures, as well as to include captions for the acronyms. Tables and figures should be self-explanatory, without requiring the readers to use the text for their understanding. 

References:

- It would be interesting to update the references, especially those below the year of 2014.

Reviewer #2: This cross sectional study aims to test the hypothesis that poverty-alleviating policies can reduce child mortality, by examining the association between the Brazilian Bolsa Familia conditional cash transfer program and child mortality (1-4 years) as well as exploring heterogeneous effects by causes of death, maternal race, gestational age, municipality income level and indexes of quality of BFP management. 

Comments:

The RECORD statement checklist of items, extended from the STROBE statement, that should be reported in observational studies using routinely collected health data is provided within the supplementary documentation.

"In fact, Bolsa Família is not randomly assigned. Instead, it is a process of families' 'self-selection' since whether a family receives the stipend or not is determined by a set of family socioeconomic characteristics. To account for the issue of self-selection bias into the exposure group, we follow a kernel matching approach for the choice of a set of PBF non-beneficiary observations inside the Cidacs' 100 Million Cohort that can allow us to compare the two groups. "

The authors have undertaken a rigorous analytical approach which supports the minimisation of potential biases in the data. 

See acronym typo in quote - which happens more than once throughout the manuscript and requires remedying.

Overall, this is a well written paper with a clearly explained, detailed, thorough, and technically appropriate methodology.

The authors have undertaken suitable subgroup analyses, accounted for bias appropriately, and acknowledged the main limitations in the discussion section for accurate and transparent interpretation of the study outcomes.

Reviewer #3: Referee report for "Conditional cash transfer program and child mortality: a study of the 100 Million Brazilian Cohort"

Main comments

This paper uses an impressive sample bringing data together from different administrative sources to analyze the relationship between exposure to Brazil's conditional cash transfer program and child mortality. As such it revisits a question on which there is existing evidence, but with a larger sample (gathering data from 2006 to 2015) covering 5 million children. While the scale is impressive, the exercise unfortunately cannot say much about the causality of the relationship that is being analyzed. And as there is existing rigorously identified evidence from large scale programs (both from quasi-experimental designs, e.g. Barham 2011 in Mexico, covering more than a 1 million children; and from large-scale experimental designs (Okeke and Abubakar 2020 in Nigeria), it is not clear the contribution is sufficient to warrant publication in PLOS Medicine. These papers also need to be recognized, as well as the wider literature in economics on the impact of CCTs on health investments.

The reasons the evidence cannot be interpreted as causal is because the assumptions underlying the Kernel matching estimations are most likely violated in this setting. Matching estimators allow for causal identification only if it is reasonable to assume that selection into the program is based on variables that are observable to the analyst. But we know from a relatively large economics literature on Bolsa Familia that exposure to the program is in part driven by political criteria/processes (e.g. de Janvry et al 2012; Brollo et al 2020; among many others), and in part related to self-reported (and therefore likely strategically biased) income. This is indeed why Bolsa is one of the few CCT programs in Latin America without a straightforward set of impact evaluation results. 

Was the sub-group analysis pre-specified? If not, why these cuts of the data and not others? Should we not be concerned with multiple hypotheses testing?

Other comments

The author reference other work on the relationship between Bolsa and child mortality, but claim they do better because they use individual data. It is never particularly clear which key questions they think they can answer better with individual data, that cannot be answered with aggregate data. It would be good to highlight this more specifically.

There is a relatively long literature on using bootstrap methods for matching estimators in economics that needs to be recognized. See Abadie and Imbens (2008, 2016) and many related papers, including on bias-corrected matching estimators (Abadie and Imbens, 2011). Also, given the sensitivity of matching estimators to strong assumptions, it is good practice to show robustness with various alternative estimators. 

References

Abadie, Alberto and Guido Imbens, 2011. Bias-Corrected Matching Estimators for Average Treatment Effects

Journal of Business and Economic Statistics, January 2011, 29(1), 1-11.

Abadie, Alberto and Guido Imbens, 2008. On the Failure of the Bootstrap for Matching Estimators Econometrica 76(6), 1537-1557.

Abadie, Alberto and Guido Imbens, 2016. Matching on the Estimated Propensity Score, Econometrica, 84(2), 781-807.

Barham, Tania, A Healthier Start: The Effect of Conditional Cash Transfers on Neonatal and Infant Mortality in Rural Mexico, Journal of Development Economics, 2011, 94(1), 74-85

Brollo, F K Kaufmann, E La Ferrara, 2020. The political economy of program enforcement: Evidence from Brazil Journal of the European Economic Association 18 (2), 750-791

de Janvry, Alain, Frederico Finan and Elisabeth Sadoulet, 2012. Local Electoral Accountability and Decentralized Program Performance, with Review of Economics and Statistics, 94(3): 672-685.

Okeke Edward and Isa S. Abubakar, 2020. "Healthcare at the beginning of life and child survival: Evidence from a cash transfer experiment in Nigeria" Journal of Development Economics, 143: 102426

[LINK]

---

## [Decision Letter · Decision Letter 2]

13 Apr 2021

Dear Dr. Ramos,

Thank you very much for submitting your manuscript "Conditional cash transfer program and child mortality, a cross-sectional analysis nested within the 100 Million Brazilian Cohort" (PMEDICINE-D-20-05781R2) for consideration in PLOS Medicine’s Special Issue: Global Child Health: From Birth to Adolescence and Beyond.

Your revised paper was evaluated by a senior editor and discussed among all the editors here. It was also discussed with the Special Issue guest editors, and re-reviewed by one of the reviewers. The reviews are appended at the bottom of this email and any accompanying reviewer attachments can be seen via the link below:

[LINK]

In light of these reviews, I am afraid that we will not be able to accept the manuscript for publication in the journal in its current form, but we would like to consider a further revised version that addresses the reviewers' and editors' comments. Obviously we cannot make any decision about publication until we have seen the revised manuscript and your response, and we plan to seek re-review by one or more of the reviewers. 

We expect to receive your revised manuscript by May 04 2021 11:59PM. Please email us (plosmedicine@plos.org) if you have any questions or concerns.

We look forward to receiving your revised manuscript. 

Sincerely,

Caitlin Moyer, Ph.D.

Associate Editor 

PLOS Medicine

plosmedicine.org

1. Reviewer Comments: Please fully address the comments of Reviewer 3.

2. Abstract: Line 39, and throughout: Please capitalize Black.

3. Author summary: Line 62 (and throughout): Please use “low-middle income countries” or similar rather than “developing” countries.

4. Introduction: Line 127-128: Please revise to avoid causal implications: “To address this gap, we tested the hypothesis that receiving BFP is associated with reduced child mortality…”

5. Methods: Please explicitly state in the Methods whether or not your study had a prospectively developed analysis plan. At line 276-278, please refer to the supporting information file containing your protocol (S1_Protocol). Please clarify if the analyses/outcomes described here are prospectively described in the included protocol. In either case, any changes in the analysis-- including those made in response to peer review comments-- should be identified as such in the Methods section of the paper, with rationale.

6. Results: Please provide statistics/results to accompany the statements comparing the strength of associations between subgroup associations, for example where associations between BFP and mortality are reported to be stronger for preterm compared to associations for term children: “Considering gestational age, an association between BFP was found for both term (Weighted OR=0.84; CI 95%: 0.79 to 0.89, p<0.0001) and preterm (weighted OR=0.78; CI 95%: 0.68 to 0.90, p<0.0001) children, with stronger association for the latter.” Please also complete this for the DMI quintile and income associations in the previous paragraphs.

7. Discussion: Lines 425-429: Would you please explain further regarding the potential for applying a regression discontinuity design in your study? It would be ideal to conduct this analysis in the current study, if feasible.

8. Discussion: Between the Limitations and Conclusions sections, please include a short section summarizing some implications and next steps for research, clinical practice, and/or public policy;

9. Discussion: Lines 441-443: Please temper causal language throughout, for example, in the following sentence: “...we can also conclude that when properly administered these programmes can increase their already significant impact on child health outcomes.”

10. Funding, Competing Interests, Data Sharing: Page 14: Please remove these sections from the main text. This information should be accurately and completely entered in the manuscript submission form.

11. Patient consent, Page 14: Please remove this section and ensure the relevant participant consent information is noted in the Methods section.

12. Table 1: Please present numbers in addition to percentages.

13. Table 3: Please define all abbreviations used in the table.

14. Figure 1: Please define all abbreviations in a figure legend (PBF under “exclusions”)

15. RECORD Checklist: Thank you for including the checklist. Please do remove references to page numbers. Locations can be referred to by section, and paragraph within section, such as: “Title Page” or “Methods, 2nd paragraph” for example.

16. Supporting Information Figures and Tables: Please provide a title and legend for each Table and Figure, and please define all abbreviations used in the figures and tables. Please report p values as p<0.001 rather than p<0.000 where applicable. Please define the asterisks (*) in Table S2a.

Comments from the reviewers:

Reviewer #3: I appreciate the efforts of the authors to revise the manuscript, and in particular to tone down the causal language throughout the manuscript. This indeed is appropriate. The revision however points to one new first order concern.

In particular, the results in Table S1c clearly show that for municipalities for which 90% or more of records are submitted, there is no significant relationship between Bolsa and child mortality. This seems to imply that possibly all the results in the paper are driven by sample selection related to missing death records, rather than any real associations between the program and child mortality (as for the subsample for which selection is small, no significant association is found. Most notably, no significative associations are found in municipalities with the full (100%) or almost full (95%) records, and indeed the results there have the opposite sign). On line 431 the authors suggest that loss of records could lead to a small overestimate. This is however not a small overestimate - as the entire relationship becomes insignificant or indeed reversed! This is particularly a concern as a very large share of the observations (3.8 million) come from municipalities with 50% of records missing. At the very least, a bounding exercise based on reasonable assumptions about the missing records would seem important, though given the magnitude of the selection, this is likely to result in very wide confidence intervals, which would unfortunately confirm that little can be said from this incomplete data about the true relationship.

In addition, the revision I continue to have many concerns with this revised manuscript that are similar to the ones raised earlier, but also with the responses to the authors to my earlier comments, which I do not believe fully address the central points I made.

In particular

1) In order to demonstrate the contribution of this paper to the literature, it is crucial not only to review what it is known from the international literature about CCTs and health investments, and what is known about Bolsa Familia in particular, but also specifically to refer to studies in the literature that causally identify the impact of this type of program on child mortality. The current approach in this paper, which refers to the international literature regarding broader health findings, but specifically excludes the papers focused on child mortality (including Barham, 2011 and Okeke and Abubakar, 2020 that I cited earlier), is highly misleading, and frankly hard to understand. I strongly suggest to acknowledge these studies, and to discuss the contribution of the current paper in comparison with them (which appears to be bringing evidence on brazil, at scale and for different subgroups, even if it is not causal).

Indeed, the concern I raised earlier is about the contribution of this paper - not about the plausibility. Given that what we already know from more rigorous designs on how these type of programs affect child mortality, why is showing associational evidence for Bolsa a sufficient contribution for publication in a high impact journal such as Plos Medecine?

2) My earlier comments on the drivers of selection into the program are NOT about suggesting a different research question - rather it points out that any study that wants to say something about "effects" (something the authors continue to claim they want to do in their response to my comments) needs to start from a good understanding of the selection criteria, given that the matching estimator proposed by the authors specifically relies on the assumption that it is observable characteristics that drive selection into the program (and that the variables used for that selection are observable to the analysist too). 

The references I provided from the economics literature are just some of many that explain (and show empirical support) that in the case of Bolsa Familia, selection is not based on the type of observables the authors account for in the propensity score estimation (which means the assumptions underlying the propensity score estimates are invalid). The concern is in part with manipulation of eligibility - either because of issues with self-reported income, or local political processes. All these capture unobserved factors driving eligibility, and therefore raise serious doubts about the assumptions underlying the matching estimations proposed by the authors (to the extent that it makes it very, very likely that the assumptions are indeed violated.). Here too, ignoring the large literature that has considered this question and instead claiming starting on page 196 that "whether a family receives the program or not is determined by a set of family socio-economic characteristics" is misleading. In this paragraph, a correct discussion on what actually drives selection into the program is important. The statement needs to be revised to give a more comprehensive review of factors affecting receipt of the program. 

Given the above, the matching estimator here merely is used as a tool to compare treated and non-treated households that are more similar in observable characteristics, something that may still be useful, but doesn't get you closer to causality. The current wording of the paper is in line with that logic, but being explicit about what the propensity score brings you in this application, and what it doesn't, would seem important to add. 

3) With regard to the point on multiple hypotheses, the concerns stand whether the different hypotheses are tested in the same regression or not. See, for instance, Young (2019), List et al (2019) for related discussions. This is exactly one of the issues that pre-analysis plans aim to address (see Olken 2015).

In addition, the main concern here is what all other potential regressions you may have run in the exploratory analysis before setting on the ones reported. All of these tests are additional hypotheses and p-values would need to be adjusted for the fact that those additional hypotheses were tested. I suggest to at least acknowledge this in the manuscript.

4) Finally, the specific suggestion made in my earlier comments regarding the matching estimations, following the references cited earlier and common practice in economics, is to show robustness of results to various alternative matching estimators (1 or 5 nearest neighbour, kernel, etc.. ). This will help to demonstrate the robustness (or not) of your results.

Other comments 

Line 218: you motivate not including self-reported income in the propensity score because it would violate the exclusion restriction of the instrumental variable approach but you are not using an IV approach! The assumptions needed for IV are different than for PSM. Matching estimators rely on the assumption that it is possible to account for selection on observables that drive program participation (independently on whether they also affect the outcome or not), so that one can compare 2 groups that according to those observables have equal likelihood of participation (but some for an exogenous reason participated and others didn't). See earlier provided references. Including self-reported income in the estimations gets you closer (in this case) to the assumptions needed for PSM. Therefore, I encourage the authors to show how results change when including self-reported income in the propensity score. 

There is a similar confusion on line 426: IV and RDD are two separate quasi-experimental methods, each with separate sets of assumptions (even if discontinuity in eligibility criteria can sometimes be used as instrument, there are many other types of instruments). In the case of Bolsa, given the evidence of manipulation of income (which would be the running var in RDD estimate), assumptions needed to use a RDD estimation are probably not valid. I hence suggest taking out the reference to RDD. 

References

List, J., A. Shaikh, and Y. Xu, (2019), "Multiple Hypothesis Testing in Experimental Economics," Experimental Economics (22): 773-793.

Young, A., (2019) "Channelling Fisher: Randomization Tests and the Statistical Insignificance of Seemingly Significant Experimental Results," Quarterly Journal of Economics, 134(2).

Olken, B., (2015), "Promises and Perils of Pre-Analysis Plans," Journal of Economic Perspectives, Vol 29(3): 61-80.

[LINK]

---

## [Editor Report · Decision Letter 3]

14 Jul 2021

Dear Dr. Ramos,

Thank you very much for submitting your manuscript "Conditional cash transfer program and child mortality, a cross-sectional analysis nested within the 100 Million Brazilian Cohort" (PMEDICINE-D-20-05781R3) for consideration in PLOS Medicine’s Special Issue: Global Child Health: From Birth to Adolescence and Beyond.

Your paper was evaluated by a senior editor and discussed among all the editors here. It was also discussed with an academic editor with relevant expertise. The reviews are appended at the bottom of this email and any accompanying reviewer attachments can be seen via the link below:

[LINK]

We would like to consider an additional revised version that addresses the editors' comments. Obviously we cannot make any decision about publication until we have seen the revised manuscript and your response.

In revising the manuscript for further consideration, your revisions should address the specific points made by the editors. Please also check the guidelines for revised papers at http://journals.plos.org/plosmedicine/s/revising-your-manuscript for any that apply to your paper. In your rebuttal letter you should indicate your response to the editors' comments, the changes you have made in the manuscript, and include either an excerpt of the revised text or the location (eg: page and line number) where each change can be found. Please submit a clean version of the paper as the main article file; a version with changes marked should be uploaded as a marked up manuscript.

We expect to receive your revised manuscript by Jul 21 2021 11:59PM. Please email us (plosmedicine@plos.org) if you have any questions or concerns.

We look forward to receiving your revised manuscript. 

Sincerely,

Caitlin Moyer, Ph.D.

Associate Editor 

PLOS Medicine

plosmedicine.org

1. From the academic editor: We request that you please more thoroughly acknowledge in the manuscript the limitations pointed out by reviewer 3, regarding the missing data (records not available for linkage) and the potential for this to impact the findings and conclusions of the study.

2. Title: Please revise your title according to PLOS Medicine's style. We suggest: “Conditional cash transfer program and child mortality: A cross-sectional analysis nested within the 100 Million Brazilian Cohort”

3. Abstract: Line 24: We suggest changing to (1-4 years of age) or similar.

4. : Line 38 and line 43: Please report this as p<0.001.

5. Abstract: Line 45: Please be more specific if possible regarding the “small over-estimate” attributable to the loss of nameless death records before linkage.

6. Author summary: Why was this study done? In the first point, we suggest “reaching” rather than “completing” if appropriate.

7. Introduction: Line 112: Please replace “developed countries” with high-income countries.

8. Methods: Line 248: Please list the perinatal conditions adjusted for here (although they are also noted in Table 4).

9. Results: Throughout the Results section, please report p<0.0001 as p<0.001.

10. Results: Please include more description and interpretation of the results presented in supporting information tables S1a, S1b, and S1c.

11. Discussion: Limitations Lines 477-484: In the sentence describing the loss of nameless records (“...loss of nameless records before linkage seems to be resulting in a possible over-estimation of the association…”) it would be helpful here, and in the appropriate section of the Results, to quantify or define the over-estimation. You mention in this paragraph that the majority of observations do come from municipalities where associations were consistent with the sample as a whole. However, as brought up previously by Reviewer 3, we request that you more thoroughly acknowledge the limitation of missing data and the impact on results, and discuss the potential for this to confound the conclusions.

12. Figure 1: Please include a descriptive legend for this figure.

13. Figure 2: Please define MHDI-R and DMI in the legend. We suggest changing the y axis to something more descriptive. If possible, please present all panels on the same range of y axis values, or note the different y axis scales in the legend.

14. Table 3: Please define the abbreviation (MHDI-R) and please describe in the legend the factors that distinguish each of the models.

15. Table 4: Please define the abbreviation DMI in the legend, and please describe the factors that distinguish each of the models.

16. Tables 5 and 6: Please describe the factors that distinguish each of the models.

17. Study Protocol: We suggest renaming the file to “S1_Protocol” and double checking to be sure all information is intended to be included (e.g. the contact information on the first page, and the investigator CVs at the end of the document).

18. RECORD Checklist: Please replace the location of the Background rationale (currently noted as pages 3 and 4) with the appropriate section/paragraph of the main text (for example, within the Introduction).

19. Figure S1a: Please include a legend, with all abbreviations used within the figure defined.

20. Figure S2: We suggest revising “treated” and “untreated” to “beneficiaries” and “non-beneficiaries” respectively.

[LINK]

---

## [Editor Report · Decision Letter 4]

4 Aug 2021

Dear Dr. Ramos,

Thank you very much for re-submitting your manuscript "Conditional cash transfer program and child mortality: A cross-sectional analysis nested within the 100 Million Brazilian Cohort" (PMEDICINE-D-20-05781R4) for consideration in PLOS Medicine’s Special Issue: Global Child Health: From Birth to Adolescence and Beyond.

I have discussed the paper with my colleagues and the academic editor. I am pleased to say that provided the remaining editorial and production issues are dealt with we are planning to accept the paper for publication in the journal.

[LINK]

We look forward to receiving the revised manuscript by Aug 11 2021 11:59PM.   

Sincerely,

Caitlin Moyer, Ph.D.

Associate Editor 

PLOS Medicine

plosmedicine.org

Requests from Editors:

1. Data availability statement: Please check that the DOI for the data is accurate/working. At this time, following the link leads to an error message (“Error: Cannot connect to server”).

2. Throughout: Please carefully check the grammar and language throughout the text for errors.

3. Abstract: Introduction: At line 25, we suggest replacing “gestational age” with “preterm birth” if this helps to clarify the objective. At line 24-26, we suggest revising to “...also examining how this association differs by maternal race/skin color, preterm birth, municipality income level, and indexes of quality of BFP management.” or similar.

4. Abstract: Methods and Findings: Line 38: Please remove the extra space in the reported p value.

5. Abstract: Methods and Findings: Line 38-39: Please also provide weighted OR, 95% CI and p values for the BFP-mortality associations for preterm birth and children of Black mothers, as is reported for municipality income status and BFP management status.

6. Abstract: Methods and Findings: Line 44-45: Please remove the word “small” and revise the sentence to: “Furthermore, sensitivity analyses showed that loss of nameless death records before linkage may have resulted in an over-estimate of the measured associations between BFP and mortality…”

7. Author summary: Why was this study done? In the first point, we suggest adding “globally” if this is meant. In the third bullet point, we suggest revising to: “...the world’s largest conditional cash transfer program)...” if this is accurate.

8. Author summary: What did the researchers do and find? In the second point, we suggest revising to: “...gestational age at birth…” or similar.

9. Author summary: What do these findings mean? In the first point, please revise to “Our findings are consistent with previous studies…” or similar. We suggest revising the wording of the second point to: “The greater association among more vulnerable groups of children suggests conditional cash transfers may help to promote equity, with stronger results among those more in need.”

10. Introduction: In the first paragraph, please make sure that all statements/reported statistics are appropriately referenced with citations.

11. Introduction: Line 112: Please revise to: “...they are also present in countries including....”

12. Introduction: Line 115: We suggest changing “speedily reached” to “...rapidly implemented throughout the country…”

13. Introduction: Line 139: Please clarify to “gestational age at birth” if this is meant. It may be more straightforward throughout to refer to preterm birth status as opposed to gestational age, as gestational age was categorized into preterm or term births in your analyses.

14. Introduction: Line 139-140: Please move this sentence to the Methods, where you describe your analysis plan: “The subgroup analysis was established by our research protocol (see S4. Research Protocol, objective 3).” - this seems to be approximately lines 211-214.

15. Methods: Line 164: Please clarify which supporting information file you are referring to with “supplementary material 1” as this is not clear from the supporting information file names.

16. Methods: Line 198: We suggest changing “before the child’s 5th anniversary…” to “prior to the child reaching 5 years of age” or similar.

17. Methods: 203: Please provide a more complete reference (or a numbered reference from the References list) for “Bolsa Família. Perguntas Frequentes” - we note that the embedded link does not work.

18. Methods: Line 223: Please note how race/ethnicity was defined/reported and by whom?

19. Methods: Line 261: We suggest using “As outlined in our research protocol…” rather than “forseen” in this sentence.

20. Methods: Line 287: Please group all of these references within one set of brackets, if appropriate ([20,21,22-25,26,27]).

21. Methods: Line 293: Please change “will be based” to “was based” in this sentence.

22. Results: Line 370-372: Please avoid using italics for emphasis.

23. Discussion: Line 409: Please revise the first sentence to “We found that receiving Bolsa Família is significantly associated with a reduced probability of deaths…”

24. Discussion: Line 412: Please revise the sentence to “These findings are consistent with previous studies…”

25. Discussion: Line 417: Please revise the sentence to “We believe that interpreting individual-level findings…” or similar.

26. Discussion: Line 431: Please revise to “...an in access to health care in Brazil…” if this is meant.

27. Discussion: Line 435: Please change “premature” to “preterm” in this sentence.

28. Discussion: Line 461: Please add “...to the best of our knowledge” to the end of the sentence, or similar: “...making this the largest study on CCTs and child mortality to date.”

29. Discussion: Line 466: Please clarify the heterogeneities that you are referring to here: “We have also accounted for heterogeneities not yet assessed regarding the association of BFP with child health outcomes.”

30. Discussion: Line 471: Please change “once” to “because” or similarly clarify what is meant here.

31. Discussion: Line 485: Please remove the hyphen from “evidence-based” in this sentence, it seems it should read “...providing evidence based on a large population…”

32. Discussion: Line 486-490: We suggest revising to: “Limitations also arise from the linkage process, as loss of nameless records before linkage may have resulted in a possible over-estimation of the association between the program and child mortality, since we observed that in the municipalities without such losses, the direction of the measured association changed, and the lower bound of the confidence interval is very close to the null value of 1.00.”

33. Discussion: Line 500: Please change “superior to” to “greater than 40%” in this sentence.

34. References: Please check if reference 29 and 30 are duplicates.

35. Table 2: Please include the definition for the abbreviation BFP in the legend or elsewhere in the table.

36. Figure 2: We suggest including an X axis label for panel C such as “Maternal race” or similar, and revising “Whites” to “White” and “Blacks” to “Black” along the labels. For panel D, we suggest including a note in the legend that Term is defined as greater than or equal to 37 weeks gestation, and Preterm is defined as less than 37 weeks gestation.

37. Supporting information: The purpose of the file “Export_20210505_25908PM” is unclear, we suggest removing.

38. Supporting information figure S3a: Please define the abbreviation PBF / please clarify if this should be BFP.

39. Tables S2a and S2b: Please check if the commas should be decimal points, “88,00” for example.

40. Supporting information file: It might be helpful to separate this document into “S2 Supporting information” and “S3 Supporting Information” to make access easier.

[LINK]

---

## [Editor Report · Decision Letter 5]

11 Aug 2021

Dear Dr Ramos, 

On behalf of my colleagues and the Academic Editor, Zulfiqar A. Bhutta, I am pleased to inform you that we have agreed to publish your manuscript "Conditional cash transfer program and child mortality: A cross-sectional analysis nested within the 100 Million Brazilian Cohort" (PMEDICINE-D-20-05781R5) in PLOS Medicine’s Special Issue: Global Child Health: From Birth to Adolescence and Beyond.

Please also complete the following editorial request:

Abstract: Line 25: Please change “(term x preterm)” to “(full term or preterm birth)” if that is intended.

PRESS

Sincerely, 

Caitlin Moyer, Ph.D. 

Associate Editor 

PLOS Medicine